# The Innate Immune Response in DENV- and CHIKV-Infected Placentas and the Consequences for the Fetuses: A Minireview

**DOI:** 10.3390/v15091885

**Published:** 2023-09-06

**Authors:** Felipe de Andrade Vieira Alves, Priscila Conrado Guerra Nunes, Laíza Vianna Arruda, Natália Gedeão Salomão, Kíssila Rabelo

**Affiliations:** 1Laboratório de Ultraestrutura e Biologia Tecidual, Universidade do Estado do Rio de Janeiro/UERJ, Rio de Janeiro 20550170, RJ, Brazil; favalves@id.uff.br (F.d.A.V.A.); viannalaiza3@gmail.com (L.V.A.); 2Laboratório Interdisciplinar de Pesquisas Médicas, Instituto Oswaldo Cruz, Fundação Oswaldo Cruz, Rio de Janeiro 21040900, RJ, Brazil; 3Laboratório de Imunologia Viral, Instituto Oswaldo Cruz, Fundação Oswaldo Cruz, Rio de Janeiro 21040900, RJ, Brazil; priscila.nunes87@gmail.com

**Keywords:** innate immunity, dengue, chikungunya, immune cells, interferon

## Abstract

Dengue virus (DENV) and chikungunya (CHIKV) are arthropod-borne viruses belonging to the *Flaviviridae* and *Togaviridae* families, respectively. Infection by both viruses can lead to a mild indistinct fever or even lead to more severe forms of the diseases, which are characterized by a generalized inflammatory state and multiorgan involvement. Infected mothers are considered a high-risk group due to their immunosuppressed state and the possibility of vertical transmission. Thereby, infection by arboviruses during pregnancy portrays a major public health concern, especially in countries where epidemics of both diseases are regular and public health policies are left aside. Placental involvement during both infections has been already described and the presence of either DENV or CHIKV has been observed in constituent cells of the placenta. In spite of that, there is little knowledge regarding the intrinsic earlier immunological mechanisms that are developed by placental cells in response to infection by both arboviruses. Here, we approach some of the current information available in the literature about the exacerbated presence of cells involved in the innate immune defense of the placenta during DENV and CHIKV infections.

## 1. Introduction

Over the last 40 years, the emergence and re-emergence of dengue have posed a considerable threat to global health, with the last 10 years seeing consecutive outbreaks of the equally severe chikungunya [1]. Throughout these years, many studies have been conducted in order to understand infection control, pathogenesis, and the host immune response to these diseases and much has evolved in this knowledge [2,3,4,5,6]. In light of this information, for a long time, there has been a considerable gap in the knowledge and understanding of these infections in pregnant patients, namely, about the claim that there really is vertical transmission, the effects on the development of the pregnancy and fetus, and the immunological effects of these infections. We know that the innate immune response plays an extremely relevant role in viral infections, acting systemically and, also, locally [4,5,6]. Thus, in this review, we will investigate the already-known aspects of the innate response to these infections in a specific organ, the placenta, in order to compile and better clarify its role in the consequences and resolution of the infection.

### 1.1. The Dengue Virus 

Although the history of dengue is uncertain, the earlier registers of a disease consistent with dengue fever date back to the period of the Chinese dynasty, on the territory of the present-day People’s Republic of China [7]. Later on, between 1779 and 1780, the illness affected the continents of Africa, Asia, and North America, causing the first well-known epidemics of dengue [8]. In spite of that, the isolation of the dengue virus was performed in 1943. Between this period and nowadays, large outbreaks occurred worldwide [2,9,10].

The etiological agent of the disease, dengue virus (DENV), is an arthropod-borne virus (arbovirus) belonging to the *Flaviviridae* family and *Flavivirus* genus, comprising four major antigenically distinct serotypes (DENV 1–4), each one capable of causing the sickness [11]. All serotypes circulate mostly in tropical and subtropical areas of the globe due to the temperature and rainy seasons, factors that are favorable to the life cycle of mosquitoes of the genus *Aedes*, remarkable vectors of arboviruses [12,13]. According to the World Health Organization (WHO), it is estimated that 25,000 deaths occur per year and over 2 billion people live in endemic areas [14].

DENV, which shares similarities with other flaviviruses, such as Zika virus (ZIKV), Japanese encephalitis virus (JEV), and yellow fever virus (YFV), is an icosahedral enveloped virus of approximately 40–50 nm in size, composed of a lipid bilayer where the structural proteins of the membrane (M) and envelope (E) are inserted [15]. Inside the lipid bilayer, there is the nucleocapsid (N), a structure composed of the viral genome surrounded by multiple copies of the capsid protein (C) [15,16,17,18]. The virus genome consists of a single positive-strand RNA of about ~11 kb in length with a 5′ cap end and lack of polyadenylated tail at its 3′ end. This genome has only one open reading frame (ORF) that is translated into a single large polyprotein that, later on, is cleaved by cellular and viral proteases in another ten distinct proteins: three structural proteins (C, prM, and E) that constitute the viral particle and seven non-structural proteins (NS1, NS2A, NS2B, NS3, NS4A, NS4B, and NS5) related to both the viral replication process and the assembly of the virions [18,19,20].

### 1.2. The Chikungunya Virus 

Chikungunya virus (CHIKV) is an arbovirus that belongs to the *Togaviridae* family and *Alphavirus* genus. It is classified as Old-World alphavirus, due to its geographical origin, and is more associated with the predominance of polyarthralgia [21,22]. Its first isolation was in 1953 in Tanzania (East Africa); it was obtained from a fevered man’s blood once it was found to be responsible for causing a febrile illness known as chikungunya fever (CHIKF) [23]. Since then, the virus has been identified in more than 60 countries in Asia, Africa, Europe, and the Americas. As of now, three genotypes have been identified: West African, East-Central-Southern African (ECSA) and Asian, and the Indian Ocean lineage, originating from ECSA [24].

The CHIKV genome is a single-strand positive-sense RNA molecule, with 11.8-kbp, encoding 2472 amino acid nonstructural and 1244 amino acid structural polyproteins [25], which gives rise to four nonstructural (nsP1-4) and five structural proteins (C, E3, E2, 6K, and E1) [26,27]. The nonstructural proteins are responsible for the viral replication; meanwhile, structural proteins shape the viral particle with a 60–70 nm diameter, which is enveloped with icosahedral nucleocapsid [28]. 

## 2. Transmission and Clinical Manifestations

### 2.1. In Dengue 

The transmission of dengue occurs through the bites of female hematophagous mosquitoes of the genus *Aedes*, mainly *Aedes aegypti*; although, other species, such as *Aedes albopictus*, are also important vectors of the disease [7,29]. Expansion of these vectors, especially *Aedes aegypti*, which is more adapted to the urban environment, is in close association with the exponential increase in urbanization, climatic changes, and socio-economic factors [30]. Transplacental transmission, organ transplantation, and blood transfusion are also types of dengue transmission reported in the literature; although, they are rare and unusual [31,32,33].

At the end of the 1990s, dengue was classified according to the parameters of the WHO, which included undifferentiated fever, dengue fever (DF), dengue hemorrhagic fever (DHF), and dengue shock syndrome (SCD) [34]. In general, the incubation period of the virus lasts between 4 to 7 days and the infection by any DENV serotype can cause a wide variety of symptoms and clinical manifestations, from a mild illness with undifferentiated fever to a life-threatening hemorrhagic fever [35,36].

Dengue fever (DF) was characterized by the presence of common symptoms, such as fever, arthralgia, headache, emesis, myalgia, and cutaneous rash [35]. A small number of patients tended to progress to a more severe clinical condition called dengue hemorrhagic fever (DHF), in which hemorrhagic manifestations, homeostasis abnormalities, and increased vascular permeability features could be noticed. Therefore, DHF was classified into four degrees of severity, with the latter (III and IV) coinciding with dengue shock syndrome (SCD), characterized by hypovolemic shock, with slight arterial pulse and hypotension [34]. However, the criteria used were outdated for applying during large outbreaks and difficult to meet, which led the WHO to create a new classification scheme [37]. This new consensus introduced the concept of classifying dengue into dengue without warning signs, dengue with warning signs, and severe dengue [14].

In dengue without warning signs, symptoms such as rash, nausea, vomiting, myalgia, arthralgia, and leukopenia, among others are common; meanwhile, the warning signs include abdominal pain, persistent vomiting, accumulation of fluid in the cavities, mucosal bleeding, and liver enlargement. Usually, hepatomegaly precedes plasma leakage, being an indicator of the evolution of the severity of the disease. On the other hand, clinical manifestations characteristic of severe dengue include severe plasma leakage, severe hemorrhage, and severe organ involvement [14,38,39].

The liver appears to be the central target during dengue infections and its involvement seems to be a usual complication [40]. This is supported by the presence of the dengue virus in this organ already being demonstrated in several studies, as well as hepatic injury due to the infection [41,42,43,44,45]. In contrast, atypical manifestations during infection, such as the commitment of the central nervous and skeletal muscle systems, heart, and lungs were also reported [46,47,48]. Nonetheless, previous studies showed the presence of the virus in the kidneys, pancreas, spleen, and even the placenta, which are unusual sites of the infection [46,49,50,51,52]. Thus, today we consider dengue to be a broad disease that affects the entire body and can cause systemic damage.

### 2.2. In Chikungunya 

CHIKV is transmitted mainly by infected mosquitoes from the *Aedes* species, such as *Aedes aegypti* and *Aedes albopictus*, and is prevalent in urban and peri-urban areas, respectively [53]. The virus had already been detected in semen and vaginal secretions; however, sexual transmission was not confirmed [54]. Usually, the first infection occurs in the skin: in fibroblast, keratinocytes, and endothelial cells from blood vessels. Upon reaching the bloodstream, the virus disseminates to various organs, such as the lymphoid tissues, liver, muscle, spleen, heart, and brain [55].

The incubation period, which is the time between infection and the onset of symptoms, lasts between 3 to 12 days. In the acute phase, the most common symptoms are high fever (>38.5 °C), rash, and intense polyarthralgia; this gives the name of the disease, which originated from the Makonde language meaning “that which bends up” [56], due to hunched posture of infected individuals and it being a disease of high morbidity [57,58]. In addition, headache, discomfort in the throat, abdominal pain, constipation or diarrhea, persistent conjunctivitis, vomiting, and lymphadenopathy (cervical or generalized) may also occur [59]. It is not rare to observe dermal manifestations, mainly on the face, trunk, and extremities [60]. A maculopapular rash is the most common cutaneous manifestation in adults and vesiculobullous lesions are predominant in children [61,62,63,64,65]. During the post-acute phase, individuals may present with arthritis; rheumatic disorders, such as tenosynovitis; bursitis; enthesitis; periostitis; and tendonitis. Clinical manifestations could persist, evolving into a chronic disease for months or years, including joint pain and swelling varying in intensity and frequency [66].

Although it is not common, some individuals develop severe forms of the disease, with multiple organ dysfunction characterized by vascular congestion, edema, and hemorrhage [67] or culminating in death. Atypical manifestations, such as respiratory disorders, arterial hypertension, hepatitis, myocarditis with sinus tachycardia, cardiomegaly, ectopic ventricular beats, abnormal electrocardiograms, and congestive heart failure, were reported [68,69,70]. Age and comorbidities (such as diabetes; cardiovascular, respiratory, renal, and autoimmune diseases; and hypertension) seem to be important factors for such; however, they may occur in low-risk populations [67,68,71]. Regarding asymptomatic individuals, the percentage is between 3 and 28% [72]. A total of 123,000 severe cases of CHIKV infection were reported in an important outbreak in the 2005–2006 period located on Reunion Island, in which about a third of the population was affected [68,71,73]. It was associated with the E1-A226V mutation, a single nucleotide change at E1 glycoprotein position 226 of the ECSA genotype resulting in an alanine (Ala) to a valine (Val) substitution. This mutation was identified in more than 90% of the isolates in the Reunion Island outbreak [26]; it seems to improve CHIKV infectivity and replication in *Aedes albopictus* and, consequently, its dissemination to humans [74]. 

## 3. The Placenta 

Previous evidence of DENV and CHIKV outbreaks has demonstrated that pregnant women are at high risk of experiencing pregnancy complications during viral infection [52,75,76,77,78]. In addition, there are some reports of the vertical transmission of these microorganisms, raising awareness of the importance of better understanding the role of the placenta in DENV and CHIKV infections [79,80,81]. Established in the third week of gestation, the placenta is characterized as a temporary and chimerical organ, formed by maternal and fetal tissue, that plays an essential role in the development and support of pregnancy. This organ supplies essential oxygen, nutrients, and hormones to the fetus, as well as carrying out the elimination of toxic waste [82].

The maternal portion of the placenta is called the decidua basalis, a tissue derived from the endometrium. On the other hand, the fetal portion includes several types of embryo-derived trophoblastic cells. These cells are specialized epithelial cells that are essential for the establishment and continuation of pregnancy. The fetal portion projects the chorionic villi, the functional unit of the placenta. They are characterized as an arboreal structure that can be anchored in the decidua or float in the intervillous space. The villi have an apical layer of syncytiotrophoblasts, which comprises the first barrier of placental defense against invading pathogens, followed by a layer of cytotrophoblastic progenitor cells and villous stroma that contain stromal fibroblasts, Hofbauer cells, and fetal vascular endothelium cells [83,84]. From the second semester, the chorionic villi are bathed by maternal blood, derived from vessels of the decidua basalis, in the intervillous space. Therefore, the human placenta is said to be hemochorial, meaning maternal blood is in contact with trophoblastic cells of fetal origin [85]. 

In this way, maternal and fetal blood do not mix, except for the rupture of capillary walls, which rarely occurs outside of the delivery situation. The separation between fetal and maternal blood is called the placental barrier, which is composed of syncytiotrophoblast, cytotrophoblast, connective tissue (containing mesenchymal cells and fibroblasts), and fetal endothelium. However, as pregnancy advances, the cytotrophoblast layer thins and disperses, making the placental barrier thinner, optimizing the exchange of substances [85].

## 4. Placental Immune Cells 

The proper development of a pregnancy requires a series of physiological adaptations and a highly dynamic balance in the maternal immune response [86,87]. This is because the fetus and placenta consist of a semi-allogeneic graft and, for this reason, adaptations are necessary in the maternal immune system, which is aimed at immune regulation and fetal tolerance parallel to an effective immune defense [88]. So, maternal immune cells are subject to constant modifications in subpopulations [89], with the upregulation of those involved with innate immunity [90].

In early pregnancy, the pro-inflammatory environment, rich in dendritic cells and natural killer (NK) cells, supports tissue remodeling and trophoblastic invasion, essential for placental establishment [91]. Natural killer cells make up about 70% of decidual leukocytes in early pregnancy [92]. These cells contain a distinct phenotype of peripheral natural killer cells and secrete several growth factors, as well as angiogenic factors and cytokines that contribute to remodeling the decidua and spiral arteries [93,94]. On the other hand, dendritic cells make up only 2% of decidual leukocytes and participate in the early stages of implantation by secreting stromal cell-derived factor 1 (SDF-1), which aids in vascular expansion and decidual angiogenesis [92,95].

In addition, the decidual immunity cell population is also composed of decidual macrophages (20–25%) [92]. Decidual macrophages are the major antigen-presenting cells (APCs) at the maternal–fetal interface in early gestation; these cells are thought to also participate in vascular remodeling, trophoblastic invasion, and immune tolerance [96,97,98,99]. Most decidual leukocytes are recruited primarily by chemokines, such as CXCL12, CXCL8, TGF-β, and CCL2, secreted by trophoblast cells and decidual cells [85,92].

As pregnancy advances, placental growth slows and the peripheral environment becomes anti-inflammatory, with Hofbauer cells and regulatory T cells secreting anti-inflammatory cytokines that aid fetal immune tolerance and rapid fetal growth [91]. In general, it can be said that fetal immune tolerance is regulated by the restriction and modulation of some leukocytes present in the maternal–fetal interface. Despite the high density of natural killer cells, the number of dendritic cells and effector T cells is relatively small. In addition, the dendritic cells present in the decidua have a unique behavior: after exposure to the fetal antigen, these cells are retained in the decidual stroma and, therefore, are not able to migrate toward the maternal lymphatic vessels [88,91]. Thus, fetal antigens reach maternal lymph nodes only by passive transport and are presented to T cells by lymph-node-resident dendritic cells, a paradigm that does not trigger an effective immune response [88,100].

In the last stage of pregnancy, the maternal immune system shifts again to a pro-inflammatory state that will be essential at the time of delivery since the uterine musculature will have to contract and expel the fetus in addition to releasing the placenta [101].

The innate immune response is responsible for controlling the viral spread during the early stages of infection [102]. The effectiveness of the innate immune system is especially important during pregnancy since vertical viral transmission can lead to developmental anomalies, intrauterine growth restriction, and premature delivery/stillbirth [90]. The role of decidual innate immune cells in the defense against viral infections and their role in vertical transmission is an emerging field; but, it is still little explored. Later, we will discuss what is known about the involvement of these cells during viral infection by DENV and CHIKV. 

## 5. Vertical Transmission in Dengue 

Despite the high incidence of the disease, studies related to the maternal/fetal consequences of DENV infection during pregnancy are still limited. In addition, there is still no consensus regarding the effects of the infection on pregnant women and/or newborns; however, some studies indicate that vertical transmission can occur and present severe outcomes, such as premature births and maternal/fetal death [103,104,105,106,107,108,109,110,111,112].

Although pregnancy is considered a risk factor for the clinical course of the disease, previous studies have not found an association between the severity of maternal infection and neonatal disease [113,114]. However, it is suggested that maternal natural immunosuppression during pregnancy may favor the occurrence of more severe infections, causing damage to the health of the mother and fetus [115]. 

In Brazil, a study carried out by Paixão et al. (2018) reported a risk of maternal death three times higher in cases of dengue and four-hundred-and-fifty times higher when the pregnant woman had DHF [107]. In addition, a study by our group showed that the severity of dengue fever led to the death of a pregnant patient, with an intense inflammation profile in the placental and fetal tissues analyzed [52]. 

A recent study in India carried out by Brar et al. (2021) observed that the average gestation period was 31.89 ± 7.31 weeks. The incidence of maternal systemic complications was high: 52.3% of pregnant women had thrombocytopenia, 25% developed postpartum hemorrhage, 18.2% of pregnant women developed acute kidney injury, 4.5% required hemodialysis support, 18.2% developed acute respiratory distress syndrome (ARDS), 15.9% required ventilatory support, 9.1% developed acute liver failure, 40.9% had evidence of shock, and 15.9% of women died. With regard to the fetus, it was observed that 4.5% of pregnancies suffered spontaneous abortion, 9% were stillbirths, and 4.5% evolved to neonatal deaths. In addition, they reported that premature babies were born in 34.1% of cases and 29.5% of women had low birth weight babies [116].

In Mexico, of the pregnant women infected with DENV in 2013, 65.9% were classified as being without warning signs of dengue (WWSD), 18.3% with warning signs of dengue (WSD), and 15.9% with severe dengue (SD). Pregnant women with SD (38.5%) had fetal distress and underwent emergency cesarean sections; this condition was associated with obstetric hemorrhage (30.8%), pre-eclampsia (15.4%), and eclampsia (7.7%). Pregnant women who did not have SD had full-term pregnancies, delivered vaginally, and had apparently healthy babies with normal birth weights [117].

In Vietnam, an investigation of pregnant women infected with DENV in 2015 showed that 90% were positive for the NS1 antigen and primary infection, 20% had premature births, and 5% had stillbirths. All neonates born alive were discharged uneventfully and no maternal death was reported [118].

During pregnancy, the fetus may be susceptible to DENV infection, especially during the critical period of organogenesis or in late pregnancy [119,120,121,122,123].

A recent study evaluated pregnant women during an epidemic in French Guiana and reported a vertical transmission rate of 18.5%, with viral transmission, both at the beginning and at the end of pregnancy. It was possible to verify that it is more frequent when maternal infection occurs late during pregnancy, close to delivery, and that newborns may present neonates with warning signs of dengue that require platelet transfusion. Furthermore, it points out that if there is a fever during the 15 days prior to delivery, the cord blood and placenta should be sampled and tested for the virus and the newborn should be closely monitored during the postpartum period [124].

Viral transmission to the fetus via the placenta can occur via the movement of the maternal vascular endothelium to trophoblasts by infected maternal monocytes, which transmit the infection to placental trophoblasts; they also do so via paracellular pathways from maternal blood to the fetal capillaries [125,126]. It has recently been reported that DENVs preferentially infect the decidua; the intensity of the decidual infection appears to be associated with the risk of fetal infection. Viral infection in the decidua in early pregnancy may modulate decidual roles in arterial remodeling and placentation that eventually influence the placental barrier balance [127].

Potential mechanisms by which a maternal infection could result in fetal death include direct fetal infection and organ damage, placental infection resulting in decreased transmission of nutrients and oxygen, and increased production of cytokines and chemokines [128].

In the histopathological evaluation of pregnant women with dengue during pregnancy carried out by Ribeiro et al., (2017), signs of hypoxia, choriodeciduitis, deciduitis, and intervillitis were observed and viral antigens were found in the trophoblast cytoplasm, villous stroma, and decidua. In this study, two possible mechanisms of fetal and neonatal morbidity were proposed: the presence of hemodynamic changes during pregnancy that could affect the placenta and cause fetal hypoxia or the direct effect of the infection on the fetus [129].

## 6. Vertical Transmission in Chikungunya 

CHIKV-infected pregnant women usually present the same clinical presentation as non-pregnant women. Basurko and collaborators carried out a study in French Guiana between June 2012 and June 2015 in which the median term of CHIKV infection was 30.7 weeks; the appearance of symptoms occurred mainly in the third trimester, with fever, arthralgia, and headache being the most common symptoms [130]. The hospitalization rate for maternal CHIKV was greater than 50%, mainly within 24 h of symptom onset; they did not observe differences in the frequency of pregnancy and neonatal outcomes when comparing to the control group (pregnant women who had no fever, no dengue, and no CHIKV infection at gestation) [130]. Similar results were found by Foeller in Grenada (August–December 2014); however, they found intense arthralgia and myalgia but with shorter durations in women who became infected with CHIKV during gestation [131]. Both authors found that the frequency of newborns who need intensive care unit admission seems to be higher when the women are exposed to CHIKV within 1 week before delivery, as well as pregnancy complications [130,131]. In contrast, a study conducted in India between August and October 2016 enrolled 150 CHIKV-infected pregnant women with a mean period of gestation of 25.62 ± 13.475 weeks. Of these women, 30 developed adverse pregnancy outcomes, mainly during the third trimester (80%), such as preterm delivery (7.33%), premature\rupture of membranes (3.33%), decreased fetal movements (2.67%), intrauterine death (2.67%), and oligohydramnios and preterm labor pains (2%) [132]. In the same way, AbdelAziem and collaborators reported cases of miscarriage (19.4%), preterm birth (13.9%), and stillbirth (4.3%) in a total of 93 women [133]. 

Although rare, vertical transmission (mother-to-child) has already been reported in CHIKV infection. The first report was conducted in June 2005 on the Reunion Island epidemic, which occurred between March 2005 and December 2006 [134]. In this outbreak, the rate of vertical transmission was close to 50% in mothers with high viremia during the intrapartum period [135]. Most authors believe the infection occurs by microtransfusions at the placental barrier or the breakdown of the syncytiotrophoblast due to uterine contractions [136,137]. The role of the placenta in CHIKV transmission is not fully understood; however, even after postponing normal birth or performing a cesarean delivery, the transmission of the virus to the baby is not avoided [135,136]. CHIKV antigens were detected in the placenta, such as the decidual, trophoblast, endothelial, Hoffbauer cells, and inside fetal capillaries [77,78,138].

During the intrapartum period, when the mother presents high viremia, the risk of the occurrence of CHIKV vertical transmission is increased; however, early maternal–fetal transmission of the virus has also been reported. Three cases of CHIKV infection before 16 weeks of gestation were reported, culminating in spontaneous abortions, with viral genome detection in the amniotic fluid, chorionic villi, and fetal brain [139]. Our group reported that spontaneous abortions occurred during the first and second trimesters, which exhibited microscopical and ultrastructural alterations and CHIKV antigen detection in abortion material [138]. The pregnant women infected with CHIKV in the studies cited were aged between 24 and 40 years old. In general, they denied smoking, alcohol use, or comorbidities. In most cases, infections in the first or second semester were symptomatic and led to miscarriage. The placentas of pregnant women who became infected with CHIKV during the second and third trimesters also exhibited histopathological alterations, CHIKV antigen detection, and an increase of cellularity and cytokines (pro- and anti-inflammatories) [77]. Several studies demonstrate the presence of CHIKV in the placenta [140], newborn cerebrospinal fluid, amniotic fluids [141], serum [79,142], and urine [79]. Although RNA CHIKV was detected in breast milk, transmission to infants was not reported [143].

Some of the obstetric complications already reported in CHIKV infection were: spontaneous abortion, preeclampsia, postpartum hemorrhage, premature birth, intrauterine death, oligohydramnios, and sepsis [76,132]. It is recommended to observe, for 7 days, the newborns of mothers who are suspected of having CHIKV infection as symptoms in infected neonates usually appear between the 3rd and 7th day of life [144]; these symptoms include fever, refusal to breastfeed, rash, swollen extremities, skin hyperpigmentation, thrombocytopenia, and irritability. However, neurological involvement may occur, leading to cases of meningoencephalitis, cerebral edema, intracranial hemorrhage, seizures, postnatal microcephaly, cerebral palsy, and neurodevelopmental delay [134,144,145,146,147,148,149]. It is important to emphasize that asymptomatic pregnant women could transmit the virus to the fetus [137]. 

## 7. Dendritic Cells, Macrophages, and Natural Killer Cells in Vertical Transmission 

Vertical transmission of the dengue and chikungunya viruses has already been shown in previous studies [33,77,114,134,138,150]. However, little is known about the intrinsic mechanisms and cells involved in this event.

Dendritic cells (DC), alongside macrophages and natural killer (NK) cells, are essential cell subpopulations in placental homeostasis, participating in the regulation of implantation events and the success of pregnancy [151]. The first ones, in particular, are abundant cells located in the basal/parietal decidua, where both CD83+ (mature dendritic cells) and DC-SIGN+ (immature dendritic cells) contribute to the homeostasis in the placental tissue and modulate the cytokine expression and function of NK cells at the maternal–fetal interface [152,153]. Furthermore, these major subpopulations of cells are considered sentinels, responsible for the dissemination and amplification of both DENV and CHIKV infection [154,155]. Even though previous works have already shown that the dendritic cells of placental tissues are permissive to ZIKV infection [156,157], in another *Flavivirus*, the exact role of these cells in the vertical transmission of DENV and CHIKV is yet to be further investigated.

Macrophages, another type of immune cell found in maternal decidua, are highly associated with several important events, including the secretion of angiogenic molecules, remodeling of spiral arteries, and clearance of apoptotic cell remains in the placental bed [158,159]. These immune cells, alongside Hofbauer cells (HC), a type of chorionic villi-resident macrophage, represent an important barrier against pathogens and play a critical role in vertical transmission [160]. Therefore, infected maternal macrophages are thought to be crucial for vertical transmission events as they could interact with the placental trophoblast cells and transmit the infection [126]. In DENV cases, Hofbauer cells and macrophages appear to be pivot cells in the pathogenesis of the disease in placental tissues as the NS3 protein, implicated in dengue virus replication, was observed in the cytoplasm of both immune cells and in several organs of aborted fetuses, as well as in the maternal and fetal region of placentas [52]. The expression of TNF-α, IFN-γ, and RANTES was also found in DENV-infected placentas, revealing the maintenance of a pro-inflammatory environment in these cases [52]. Additionally, in an immunocompromised animal model, DENV vertical transmission was observed in the early stages of pregnancy and associated with an increased antibody-dependent enhancing (ADE) condition, which makes it conceivable that Hofbauer cells and macrophages at the maternal portion expressing Fc-gamma receptors could play an important role in inducing an ADE condition and, consequently, fetal infection [161]. Regarding CHIKV infections, several virus antigens were found in Hofbauer cells in the placentas of infected pregnant women, evidencing the permissiveness of these cells to infection [78,138]. The presence of pro-inflammatory mediators was also noticed [77].

Decidual NK cells compose the majority of decidual cells (dNK) during early pregnancy and are specifically located around expanding extravillous trophoblasts [162]. These specialized maternal cells differ both in phenotype and function when compared to peripheral NK (pNK) cells and play a critical role during trophoblast invasion and placentation [163,164,165]. They also display distinct cytotoxic responses as dNK cells seem to produce high levels of cytokines and be less cytotoxic during trophoblast infection [166,167]. Therefore, dNK cells tend to preserve the placental trophoblasts during the development of an immune response against some pathogens, evidencing the fact that the placenta is considered a highly privileged organ [168,169,170,171]. Despite the fact that dNK cells are an immunotolerant subpopulation of cells, the gaining of a cytotoxic phenotype can occur regarding some specific infections [172,173].

## 8. IFN-I Response to Dengue and Chikungunya Placental Infection 

Type I interferons (IFN-I) are the main cytokine mediators of the innate immune response and constitute a key defense mechanism against viral infections [173,174,175]. Soon after a viral infection, IFN-I synthesis is rapidly induced upon detection of viral RNA by pattern recognition receptors (PRRs) and consequent activation of interferon regulatory factor (IRF) [173]. Once synthesized, these cytokines act in a paracrine fashion to induce a peripheral antiviral state [176,177]. To complete this action, different subtypes of IFN-I, including IFN beta and IFN alpha, interact with the heterodimeric IFNAR receptor (IFNAR1/FNAR 2) to trigger a JAK-STAT-mediated signaling cascade that culminates in the transcription of hundreds of genes stimulated by interferons (ISGs) that have antiviral and immunomodulatory activities [99,178]. ISG molecules can act by several mechanisms in order to repress viral replication, including the inhibition of virus entry into the cell, inhibition of viral protein synthesis, degradation of essential viral components, and changes in cell metabolism [179,180]; they even play a regulatory and immunomodulatory role [181]. As much as they act through a shared receptor, it is noteworthy that the IFN-I subtypes have different properties [175,177,181,182].

In general, it is known that DENV is capable of inhibiting IFN-I signaling by two mechanisms: directly interfering in ISG synthesis pathways in parallel with the evasion of innate immune receptors. The non-structural proteins of DENV, especially NS2, NS4 (NS4A/NS4B), and NS5, have the ability to inhibit the activation of tyrosine kinase 2 (Tyk2), inhibit the phosphorylation of STAT1, and decrease the expression and inhibit the phosphorylation of STAT 2 (essential intermediates in the ISG synthesis cascade). Furthermore, the NS5 protein induces STAT2 degradation through a mechanism involving the cellular proteasome. On the other hand, the evasion of cell receptors would be related to the site of viral replication. Like other flaviviruses, the dengue virus induces the formation of intracellular vesicles from the membrane of the endoplasmic reticulum, which functions as a viral replication site. These vesicles resemble cellular organelles and, for this reason, are not recognized by components of the innate response [183,184,185]. 

With regard to in vitro studies, Luo and collaborators performed infection tests with flaviviruses, such as ZIKV, YFV, and DENV, in first-trimester human extravillous trophoblast cells (HTR8). DENV-RNA levels in the infected HTR8 cells were significantly enhanced on day 1 and continued to increase on day 4 and day 6 pi. On day 4 pi, IL-6, TNF-α, IL-8, and CCL2 production was augmented in ZIKV-infected HTR8 cells compared to YFV and DENV; however, DENV-infected cells produced more of these cytokines compared with the YFV-infected cells. Meanwhile, CCL3 (macrophage inflammatory protein-1 α, MIP-1α) and RANTES/CCL5 production were higher in DENV-infected cells. The IFN-alpha response was low in DENV-infected cells and the IFN-beta response was higher in DENV-infected cells compared to ZIKV-infected cells in each of the three infection times; this was also the case when compared to YFV-infected cells 6dpi [186]. The cytokine profile of DENV-infected HTR8 cells was characterized by high levels of IL-6, IL-10, IL-15, CCL2, CCL3, IL-8, VEGF, IFN-gamma, and IFN-alpha 2 [187]. In addition, DENV was shown to be able to infect other trophoblast cell lines, such as JEG3 and JAR, and promote the expression of IFNλ1 better than IFNλ2 [188]. In experiments with mice infected with DENV, the decidua exhibited a higher number of genes being upregulated, including caspase (2, 6, 8, and 9), IRF1, and NOS2. In the fetal placenta, there were expressions of complements, such as C4A, C6, and CFB [161].

Although some studies have already shown that CHIKV is able to inhibit the phosphorylation of the intermediates of the JAK-STAT cascade and, therefore, interfere with the IFN-I-mediated response [189], it is already well established that the response mediated by IFN-I has a critical role in limiting the replication and pathogenesis of CHIKV in human and mouse models and that the different subtypes of IFN-I (IFN alpha and IFN beta) play a protective role via different mechanisms [174,190,191]. While IFN alpha acts by limiting viral replication and spread, IFN beta acts by modulating neutrophil density at the site of infection, regulating inflammation during acute infection [174,192]. Furthermore, it is believed that IFN alpha somehow interferes with the chronic version of the pathology. Locke and collaborators demonstrated that early IFN alpha activity is able to limit persistent viral RNA, as well as the number of surviving immune cells, suggesting that the IFN alpha-mediated response plays a central role in the development of chronic chikungunya [177]. However, further studies are needed to clarify the role of each subtype of IFN-I in the chronic condition of the pathology. 

Despite the high incidence of DENV and CHIKV infection in pregnant women, the role of IFN-I during placental viral infection is a gap in the knowledge. Studies investigating the impact of IFN-I during placental infection with DENV or CHIKV are extremely scarce. It is noteworthy that IFN-I is an essential molecule for the proper development of a pregnancy since these cytokines act in the placenta by regulating inflammation, protecting against viral infections, and contributing to fetal immunity [193,194]. Loss of an IFN-I-mediated response in the placenta can lead to a number of events, including exacerbated viral replication, fetal infection, and other factors that contribute to pregnancy complications [194,195,196]. Thus, the need for and urgency of carrying out studies evaluating the role of IFN-I in placental infection by DENV and CHIKV is evident.

## 9. Conclusions

The occurrence of arboviruses during pregnancy is an additional concern, due to the possibility of vertical transmission and fetal involvement. The various placental immune cells play a role in viral dissemination and may contribute to vertical transmission. IFN-I proteins are the main cytokine mediators of the innate immune response and constitute a key defense mechanism against viral infections. Despite the high incidence of DENV and CHIKV infections in pregnant women, the role of IFN-I during placental viral infection is a gap in knowledge and must be better studied (Figure 1). Most of the studies reported here were case studies of patients who had infections during pregnancy, some of which led to serious outcomes, such as miscarriage or maternal and fetal death. We therefore believe that infection with these arboviruses in pregnancy can be very dangerous and should be studied further. 

## Figures and Tables

**Figure 1 viruses-15-01885-f001:**
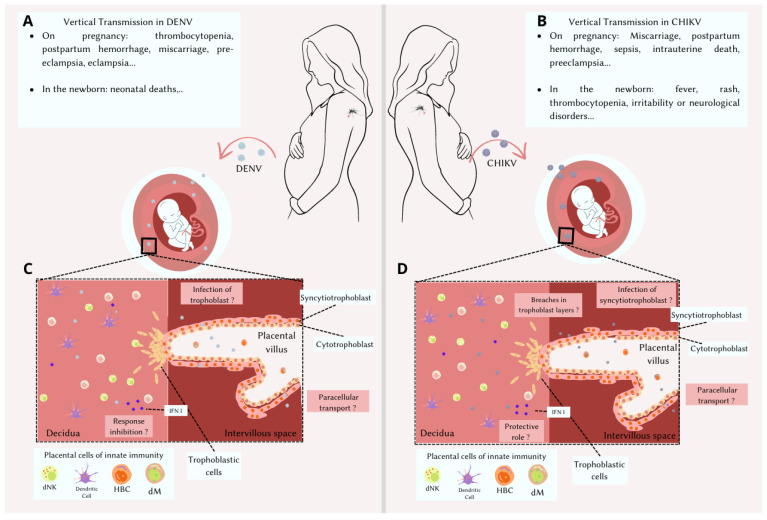
Schematic representation of the human maternal–fetal interface during DENV or CHIKV infection. DENV or CHIKV infection has immense potential to affect both maternal and fetal health. (**A**) During pregnancy, DENV infection can lead to thrombocytopenia, postpartum hemorrhage, miscarriage, and preeclampsia, in addition to representing an increased risk of neonatal death. (**B**) On the other hand, CHIKV infection can cause spontaneous abortion, postpartum hemorrhage, sepsis, intrauterine death, and preeclampsia and can also cause thrombocytopenia, fever, rash, irritability, and neurological disorders in the newborn. In the basal decidua are cells of the immune system: decidual natural killer (dNK) cells, dendritic cells, and maternal macrophages (dM). Chorionic villi contain trophoblast cells, Hofbauer cells (HBC), and fetal capillaries surrounded by a layer of cytotrophoblasts and multinucleated syncytiotrophoblast cells. The chorionic villus is floating in the intervillous space, bathed in maternal blood. So far, the mechanism involved in the vertical transmission of both viruses remains unclear. It is believed that vertical transmission can occur via the direct infection of trophoblasts (**C**) or syncytiotrophoblasts (**D**), as well as from breaches on the trophoblast layer (**D**) or via paracellular transport (**C**,**D**) from maternal blood to the fetal capillaries. The role of decidual immune system cells during DENV or CHIKV infection is not well established and nor is the IFN-I-mediated response, representing a gap in knowledge.

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
