# Peer review of "The Innate Immune Response in DENV- and CHIKV-Infected Placentas and the Consequences for the Fetuses: A Minireview"

_viruses, 2023, doi:10.3390/v15091885_

Round 1
Reviewer 1 Report
The topic addressed by the authors is interesting; however, I consider that he was missing in his writing:
1. Describe the participation of viral proteins in the known mechanisms of pathogenesis in the placenta.
2. Prepare a figure indicating the participation of the different cells of the innate immune system in the prevention of viral infections, specifically Dengue and Chikungunya viruses. They will have to explain why after making a description of the innate immune response in the placenta, they only emphasize the importance of IFN I.
3. The most frequent comorbidities in pregnant women in whom vertical transmission was observed are not mentioned. Neither, if the development of severe forms of dengue and vertical transmission are related to the dengue virus serotype, heterotypic antibodies, multiparity, age of the women, etc.
4. Finally, the title does not reflect the content of the manuscript. "Innate defense" is not necessarily indicative of the innate immune system; Furthermore, the manuscript describes the innate immune system in the placenta in the absence and in the presence of infection by both viruses; and includes a section on what happens in the fetuses and newborns of women infected with these viruses.
Author Response
Dear Reviewer,
We would like to thank you for all valuable comments on our study. They will certainly improve the quality of our final work.
The topic addressed by the authors is interesting; however, I consider that he was missing in his writing:
- Describe the participation of viral proteins in the known mechanisms of pathogenesis in the placenta.
As we reformulated the manuscript as a mini-review, we were left with a size limit on the text and therefore focused on the innate immune response, which is the topic we were invited by the journal to write about. Therefore, we were able to carry out all the other suggestions below.
- Prepare a figure indicating the participation of the different cells of the innate immune system in the prevention of viral infections, specifically Dengue and Chikungunya viruses. They will have to explain why after making a description of the innate immune response in the placenta, they only emphasize the importance of IFN I.
In fact, what we bring in the description at the end of the work is the gap in understanding the role of interferon in the situation of pregnancy. The figure depicts cells already observed to have some role in placental inflammation caused by these infections in pregnancy.
- The most frequent comorbidities in pregnant women in whom vertical transmission was observed are not mentioned. Neither, if the development of severe forms of dengue and vertical transmission are related to the dengue virus serotype, heterotypic antibodies, multiparity, age of the women, etc.
The pregnant women infected with CHIKV in the studies cited were aged between 24 and 40. In general, they denied smoking, alcohol use or comorbidities. In most cases, infection in the first or second semester led to miscarriage. In the dengue cases, the patients were young( ~20-30 years-old) and also had no comorbidities. These data have been added to the manuscript.
- Finally, the title does not reflect the content of the manuscript. "Innate defense" is not necessarily indicative of the innate immune system; Furthermore, the manuscript describes the innate immune system in the placenta in the absence and in the presence of infection by both viruses; and includes a section on what happens in the fetuses and newborns of women infected with these viruses.
We agree with what has been suggested and we have reworded the title of the work.
Reviewer 2 Report
The paper of Alves and colleagues is a review about CHIKV and DENV infection in pregnat women.
In the first chapters (which numeration should anyway be reviewed, as noted in the minor points below), the amount of information about the two viruses is insufficient,
and in particular there is too little about the DENV serotypes. Please expand with relevant information. Some figures may also help. Information about the innate immunity
aginst these viruses is not reported, and this gap should be fixed.
The chapters about placenta, on the other hand, appear to be excessively long and unnecessarily detailed in my opinion.
The following chapters are of sufficiently good quality and rather well organized, even though CHIKV looks to be sometimes left one step behind DENV.
In general terms, what I don't appreciate very much is the excess amount of references of other reviews. I know it is a general trend, but I don't think that it is a goo habit.
References to original studies should always be preferred, and I would like to see some efforts in this process in a modified version of the paper.
The real main problem with this review is that the title looks inappropriate, and the abstract misleading. Indeed, the focus should be on innate immunity in placenta during DENV and CHIKV infection,
but this topic is absent from the text. The only few lines about this are at the end of chapter 9, ando only to state that there is almost nothing to review about it in the literature, since almost
nothing has been published so far. So, the question that many future readers could pose is: why a review on a topic in which so little knowledge is available?
I think the authors should reconsider globally the organization and the scope of the paper to ensure it becomes relevant for publication
Some minor points:
- page 2, second line: "Chin dynasty" or "Qin dynasty"
- page 2, ch.3: CIKV is not classified in the old-world group due to its symptoms, but to its origins. The association with symptoms comes after
- page 2: there are two dinstinct chapters 3.
- page 3, third line: "albopictus" not "Albopictus"
- page 7: ref 125,126,127 and 128 are all from reviews. please cite some original studies
- page 8: the period starting at line 3 ("three cases...") has no verb
- page 10: the conclusion is numbered as chapter 5, but it follows chapter 9
- references: in ref 3 please check for correct title
- in ref 36 please check for correct authors names
- in ref 139 publication details are missing
I recommend a revision of complexive english language by a native speaker
Author Response
Dear Reviewer,
We would like to thank you for all valuable comments on our study. They will certainly improve the quality of our final work.
The paper of Alves and colleagues is a review about CHIKV and DENV infection in pregnant women.
In the first chapters (which numeration should anyway be reviewed, as noted in the minor points below), the amount of information about the two viruses is insufficient, 4,and in particular there is too little about the DENV serotypes. Please expand with relevant information. Some figures may also help. Information about the innate immunity
aginst these viruses is not reported, and this gap should be fixed.
The chapters about placenta, on the other hand, appear to be excessively long and unnecessarily detailed in my opinion.
The following chapters are of sufficiently good quality and rather well organized, even though CHIKV looks to be sometimes left one step behind DENV.
This article has now been rewritten as a mini-review because, as stated in the title, we want to list the studies that show aspects of innate immunity in the face of DENV and CHIKV infection in the placenta and the possible consequences for the fetus. We won't go into too much detail on the topics of dengue and CHIKV, as there are many reviews on these infections in general. The topic of the placenta is necessary because it is one of the main organs dealt with in the manuscript and its morphology has been little studied. What the referee mentions about CHIKV is a consequence of what is found in the literature, as there are many more studies on dengue than CHIKV. Even so, we've added more information to the text as we found it.
In general terms, what I don't appreciate very much is the excess amount of references of other reviews. I know it is a general trend, but I don't think that it is a goo habit.
References to original studies should always be preferred, and I would like to see some efforts in this process in a modified version of the paper.
The real main problem with this review is that the title looks inappropriate, and the abstract misleading. Indeed, the focus should be on innate immunity in placenta during DENV and CHIKV infection,
but this topic is absent from the text. The only few lines about this are at the end of chapter 9, ando only to state that there is almost nothing to review about it in the literature, since almost
nothing has been published so far. So, the question that many future readers could pose is: why a review on a topic in which so little knowledge is available?
I think the authors should reconsider globally the organization and the scope of the paper to ensure it becomes relevant for publication
We agree with what has been said about the excess of reviews in the bibliography and have rewritten it, reducing the number of reviews and increasing the number of original articles. The focus of the work is on the innate placental response to these infections, which has been described since we mentioned the organ itself. Contrary to what has been said, we don't consider only the interferon response as an innate response, but rather the role of immune cells found at the site and which are exacerbated in infections. This is well explored in the text, as there are different publications in the area and some of them include publications from our own group, which gives us expertise in the area.
Some minor points:
- page 2, second line: "Chin dynasty" or "Qin dynasty"
- page 2, ch.3: CIKV is not classified in the old-world group due to its symptoms, but to its origins. The association with symptoms comes after
- page 2: there are two dinstinct chapters 3.
- page 3, third line: "albopictus" not "Albopictus"
- page 7: ref 125,126,127 and 128 are all from reviews. please cite some original studies
- page 8: the period starting at line 3 ("three cases...") has no verb
- page 10: the conclusion is numbered as chapter 5, but it follows chapter 9
- references: in ref 3 please check for correct title
- in ref 36 please check for correct authors names
- in ref 139 publication details are missing
All the minor points were considered in order to improve the work.